# OmniControl: Control Any Joint at Any Time for Human Motion Generation

**Yiming Xie**[1]**, Varun Jampani**[2]**, Lei Zhong**[1]**, Deqing Sun**[3]**, Huaizu Jiang**[1]
[1]Northeastern University   [2]Stability AI   [3]Google Research

## Abstract

We present a novel approach named OmniControl for incorporating flexible spatial control signals into a text-conditioned human motion generation model based on the diffusion process. Unlike previous methods that can only control the pelvis trajectory, OmniControl can incorporate flexible spatial control signals over different joints at different times with only one model. Specifically, we propose analytic spatial guidance that ensures the generated motion can tightly conform to the input control signals. At the same time, realism guidance is introduced to refine all the joints to generate more coherent motion. Both the spatial and realism guidance are essential and they are highly complementary for balancing control accuracy and motion realism. By combining them, OmniControl generates motions that are realistic, coherent, and consistent with the spatial constraints. Experiments on HumanML3D and KIT-ML datasets show that OmniControl not only achieves significant improvement over state-of-the-art methods on pelvis control but also shows promising results when incorporating the constraints over other joints. Project page: `https://neu-vi.github.io/omnicontrol/`.

## 1 Introduction

We address the problem of incorporating spatial control signals over *any joint at any given time* into text-conditioned human motion generation, as shown in Fig. 1. While recent diffusion-based methods can generate diverse and realistic human motion, they cannot easily integrate flexible spatial control signals that are crucial for many applications. For example, to synthesize the motion for picking up a cup, a model should not only semantically understand "pick up" but also control the hand position to touch the cup at a specific position and time. Similarly, for navigating through a low-ceiling space, a model needs to carefully control the height of the head during a specific period to prevent collisions.

These control signals are usually provided as global locations of joints of interest in keyframes as they are hard to convey in the textual prompt. The relative human pose representations adopted by existing inpainting-based methods (Karunratanakul et al., 2023; Shafir et al., 2024; Tevet et al., 2023), however, prevent them from incorporating flexible control signals. The limitations mainly stem from the relative positions of the pelvis w.r.t. the previous frame and other joints w.r.t. the pelvis. As a result, to input the global pelvis position specified in the control signal to the keyframe, it needs to be converted to a relative location w.r.t the preceding frame. Similarly, to input positions of other joints, a conversion of the global position w.r.t. the pelvis is also required. But in both cases, the relative positions of the pelvis are non-existent or inaccurate in-between the diffusion generation process. Therefore, both Tevet et al. (2023) and Shafir et al. (2024) struggle to *handle sparse constraints on the pelvis* and *incorporate any spatial control signal on joints other than the pelvis*. Although Karunratanakul et al. (2023) introduces a two-stage model to handle the sparse control signals over the pelvis, it still faces challenges in controlling other joints.

In this work, we propose OmniControl, a novel diffusion-based human generation model that can incorporate flexible spatial control signals over any joint at any given time. Building on top of Tevet et al. (2023), OmniControl introduces spatial and realism guidance to control human motion generation. We adopt the same relative human pose representations as the model's input and output for its effectiveness. But in the spatial guidance module, unlike existing methods, we propose to convert the generated motion to global coordinates to directly compare with the input control sig-

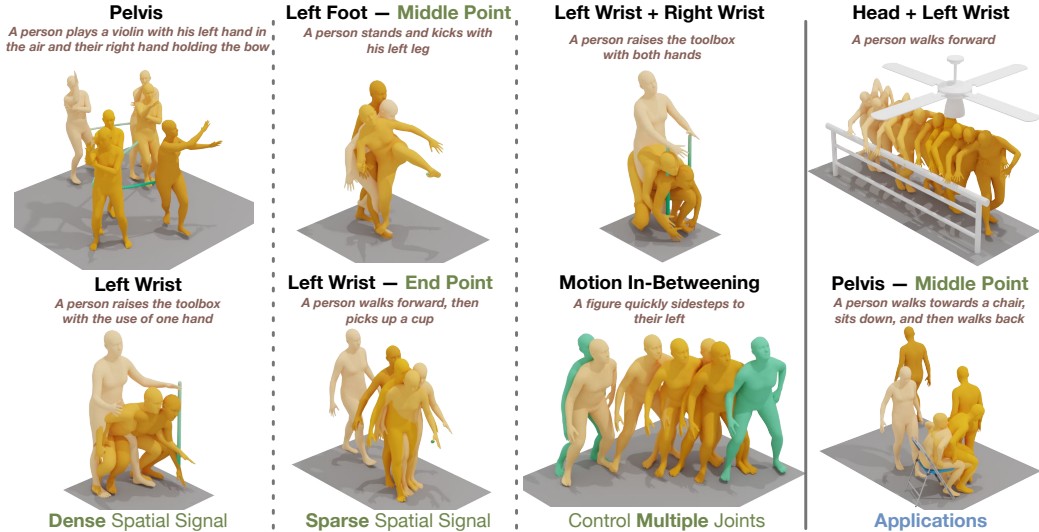

**Figure 1: OmniControl can generate realistic human motions given a text prompt and flexible spatial control signals.** *Darker color indicates later frames in the sequence. The* green *line or points indicate the input control signals. Best viewed in color.*

nals, where the gradients of the error are used to refine the generated motion. It eliminates the ambiguity related to the relative positions of the pelvis and thereby addresses the limitations of the previous inpainting-based approaches. Moreover, it allows dynamic iterative refinement of the generated motion compared with other methods, leading to better control accuracy. While effective at enforcing spatial constraints, spatial guidance alone usually leads to unnatural human motion and drifting problems. To address these issues, taking inspirations from the controllable image generation (Zhang et al., 2023b), we introduce the *realism guidance* that outputs the residuals w.r.t. the features in each attention layer of the motion diffusion model. These residuals can directly perturb the whole-body motion densely and implicitly. Both the spatial and realism guidance are essential and they are highly complimentary in balancing control accuracy and motion realism, yielding motions that are realistic, coherent, and consistent with the spatial constraints.

Experiments on HumanML3D (Guo et al., 2022a) and KIT-ML (Plappert et al., 2016) show that OmniControl outperforms the state-of-the-art text-based motion generation methods on pelvis control by large margins in terms of both motion realism and control accuracy. More importantly, OmniControl achieves impressive results in incorporating the spatial constraints over any joint at any time. In addition, we can train a single model for controlling multiple joints together instead of having an individual model for each joint, as shown in Fig. 1 (*e.g.*, both left wrist and right wrist). These proprieties of OmniControl enable many downstream applications, *e.g.*, connecting generated human motion with the surrounding objects and scenes, as demonstrated in Fig. 1 (*last column*).

To summarize, our contributions are: (1) To our best knowledge, OmniControl is the first approach capable of incorporating spatial control signals *over any joint at any time*. (2) We propose a novel control module that uses both spatial and realism guidance to effectively balance the control accuracy and motion realism in the generated motion. (3) Experiments show that OmniControl not only sets a new state of the art in controlling the pelvis but also can control any other joints using a single model in text-based motion generation, thereby enabling a set of applications in human motion generation.

## 2 RELATED WORK

### 2.1 HUMAN MOTION GENERATION

Human motion synthesis can be broadly categorized into two groups: auto-regressive methods (Rempe et al., 2021; Starke et al., 2019; 2022; Shi et al., 2023; Ling et al., 2020; Peng et al., 2021; Juravsky et al., 2022) and sequence-level methods (Tevet et al., 2023; Yan et al., 2019). Auto-regressive methods use the information from past motion to recursively generate the current motion frame by frame. These methods are primarily tailored for real-time scenarios. In contrast, sequence-

level methods are designed to generate entire fixed-length motion sequences. Owing to this inherent feature, they can seamlessly integrate with existing generative models, such as VAE (Habibie et al., 2017; Petrovich et al., 2021; Lucas et al., 2022) and diffusion models (Zhang et al., 2022a; Chen et al., 2023), enabling various prompts. These prompts can originate from various external sources, such as text (Petrovich et al., 2023; Guo et al., 2022b; Petrovich et al., 2022; Tevet et al., 2023; Chen et al., 2023; Zhang et al., 2022a; Jiang et al., 2023; Zhang et al., 2023c;a; Tevet et al., 2022; Ahuja & Morency, 2019; Guo et al., 2022a; Kim et al., 2023; Yuan et al., 2023), action (Guo et al., 2020; Petrovich et al., 2021), music (Li et al., 2022; Tseng et al., 2023; Li et al., 2021), images (Chen et al., 2022), trajectories (Kaufmann et al., 2020; Karunratanakul et al., 2023; Rempe et al., 2023), 3D scenes (Huang et al., 2023; Zhao et al., 2023; Wang et al., 2022a;b) and objects (Ghosh et al., 2023; Kulkarni et al., 2024; Jiang et al., 2022; Xu et al., 2023; Hassan et al., 2021; Starke et al., 2019; Zhang et al., 2022b; Pi et al., 2023; Li et al., 2023a).

Although incorporating spatial constraints is a fundamental feature, it remains a challenge for text-based human motion synthesis methods. An ideal method should guarantee that the produced motion closely follows the global spatial control signals, aligns with the textual semantics, and maintains fidelity. Such an approach should also be capable of controlling any joint and their combinations, as well as handling sparse control signals. PriorMDM (Shafir et al., 2024) and MDM (Tevet et al., 2023) use inpainting-based methods to input the spatial constraints into the generated motions. However, limited by their relative human pose representations where locations of other joints are defined w.r.t. to the pelvis, these methods struggle to incorporate the *global* constraints for other joints except for the pelvis and handle sparse spatial constraints. Although the inpainting-based method GMD (Karunratanakul et al., 2023) introduces a two-stage guided motion diffusion model to handle sparse control signals. it still faces challenges in incorporating spatial constraints into any other joint. In this paper, we focus on sequence-level motion generation and propose a novel method that enables control over any joint, even with sparse control signals, using a single model.

## 2.2 CONTROLLABLE DIFFUSION-BASED GENERATIVE MODEL IN IMAGE GENERATION

Recently, the diffusion-based generative model has gained significant attention due to their impressive performance in image generation (Rombach et al., 2022). Diffusion models are well-suited for controlling and conditioning. Typically, there are several methods for conditional generation. Imputation and inpainting (Choi et al., 2021; Chung et al., 2022) fill in missing parts of data with observed data such that the filled-in content is visually consistent with the surrounding area. However, it is difficult when the observed data is in a different space compared to the filling part, *e.g.*, generating images from semantic maps. Classifier guidance (Chung et al., 2022; Dhariwal & Nichol, 2021) exploits training a separate classifier to improve the conditional diffusion generation model. Classifier-free guidance (Ho & Salimans, 2021) jointly trains conditional and unconditional diffusion models and combines them to attain a trade-off between sample quality and diversity. GLIGEN (Li et al., 2023b) adds a trainable gated self-attention layer at each transformer block to absorb new grounding input. ControlNet (Zhang et al., 2023b) introduces a neural network designed to control large image diffusion models, enabling rapid adaptation to task-specific control signals with minimal data and training. These controlling methods are not mutually exclusive, and solely adopting one may not achieve the desired goals. Inspired by classifier guidance and ControlNet, we design hybrid guidance, consisting of spatial and realism guidance, to incorporate spatial control signals into human motion generation. The spatial guidance applies an analytic function to approximate a classifier, enabling multiple efficient perturbations of the generated motion. At the same time, the realism guidance uses a neural network similar to ControlNet to adjust the output to generate coherent and realistic motion. Both of these two guidance modules are essential and they are highly complimentary in balancing motion realism and control accuracy.

## 3 OMNICONTROL

In this section, we introduce our proposed OmniControl for incorporating spatial constraints into a human motion generation process. Fig. 2 shows an overview of OmniControl. Given a prompt $p$, such as text, and an additional spatial control signal $c \in \mathbb{R}^{N \times J \times 3}$, our goal is to generate a human motion sequence $x \in \mathbb{R}^{N \times D}$. $N$ is the length of the motion sequence, $J$ is number of joints, and $D$ is the dimension of human pose representations, *e.g.*, $D = 263$ in the HumanML3D (Guo et al.,

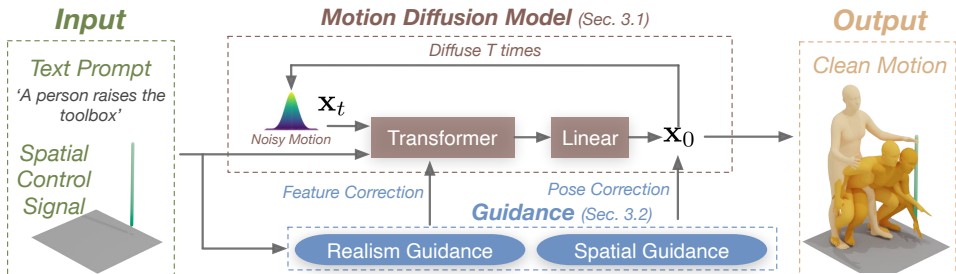

Figure 2: **Overview of OmniControl**. Our model generates human motions from the text prompt and spatial control signal. At the denoising diffusion step, the model takes the text prompt and a noised motion sequence $x_t$ as input and estimates the clean motion $x_0$. To incorporate flexible spatial control signals into the generation process, a hybrid guidance, consisting of realism and spatial guidance, is used to encourage motions to conform to the control signals while being realistic.

2022a) dataset. The spatial constraints $c$ consist of the xyz positions of each joint at each frame. In practice, only a subset of joints' locations are provided as spatial constraints for human motion generation, where the positions of the rest of the joints are simply set to zero. This form of definition enables us to flexibly specify the controlling joints and thus control the motion generation *for any joint at any time (keyframe)*. We first provide a brief overview of the text-prompt human motion generation model based on the diffusion process in Sec. 3.1. We then introduce our proposed approach to incorporate the spatial control signal $c$ into the generation model in Sec. 3.2.

## 3.1 BACKGROUND: HUMAN MOTION GENERATION WITH DIFFUSION MODELS

**Diffusion process for human motion generation.** Diffusion models have demonstrated excellent results in text-to-image generation (Rombach et al., 2022; Saharia et al., 2022; Ramesh et al., 2021). Tevet et al. (2023) extend it to the human generation to simultaneously synthesize all human poses in a motion sequence. The model learns the reversed diffusion process of gradually denoising $x_t$ starting from the pure Gaussian noise $x_T$

$$P_\theta(x_{t-1}|x_t, p) = \mathcal{N}(\mu_t(\theta), (1 - \alpha_t)I), \tag{1}$$

where $x_t \in \mathbb{R}^{N \times D}$ denotes the motion at the $t^{\text{th}}$ noising step and there are $T$ diffusion denoising steps in total. $\alpha_t \in (0, 1)$ are hyper-parameters, which should gradually decrease to 0 at later steps. Following Tevet et al. (2023), instead of predicting the noise at each diffusion step, our model directly predicts the final clean motion $x_0(\theta) = M(x_t, t, p; \theta)$[1] where $M$ is the motion generation model with parameters $\theta$. The mean $\mu_t(\theta)$ can be computed following Nichol & Dhariwal (2021) $\mu_t(\theta) = \frac{\sqrt{\bar{\alpha}_{t-1}}\beta_t}{1-\bar{\alpha}_t}x_0(\theta) + \frac{\sqrt{\alpha_t}(1-\bar{\alpha}_{t-1})}{1-\bar{\alpha}_t}x_t$, where $\beta_t = 1 - \alpha_t$ and $\bar{\alpha}_t = \prod_{s=0}^{t} \alpha_s$. We omit $\theta$ for brevity and simply use $x_0$ and $\mu_t$ in the rest of the paper. The model parameters $\theta$ are optimized to minimize the objective $\|x_0 - x_0^*\|_2^2$, where $x_0^*$ is the ground-truth human motion sequence.

**Human pose representations.** In human motion generation, the redundant data representations suggested by Guo et al. (2022a) are widely adopted (Tevet et al., 2023) which include pelvis velocity, local joint positions, velocities and rotations of other joints in the pelvis space, as well as the foot contact binary labels. Generally, the pelvis locations are represented as relative positions w.r.t. the previous frame, and the locations of other joints are defined as relative positions w.r.t. the pelvis. Such representations are easier to learn and can produce realistic human motions. However, their relative nature makes inpainting-based controlling methods (Tevet et al., 2023) struggle to *handle sparse constraints on the pelvis* and *incorporate any spatial control signal on joints other than the pelvis*. For instance, to input the global pelvis position specified in the control signal to the keyframe, the global pelvis position needs to be converted to a relative location w.r.t the preceding frame. Similarly, to input positions of other joints, such as the left hand, a conversion of the global hand position w.r.t. the relative location of pelvis is required. However, in both cases, the relative positions of the pelvis do not exist and are yet to be generated by the model. Some approaches use the generated motion in-between the diffusion process to perform the conversion to enforce the spatial constraints. But relying on the generated pelvis positions for these conversions can sometimes

---

[1]Strictly speaking, it should be written as $x_0(x_t, t, p; \theta) = M(x_t, t, p; \theta)$. So should $\mu_t(\theta)$. We slightly abuse the notations here for brevity, highlighting their dependence on the model parameters $\theta$.

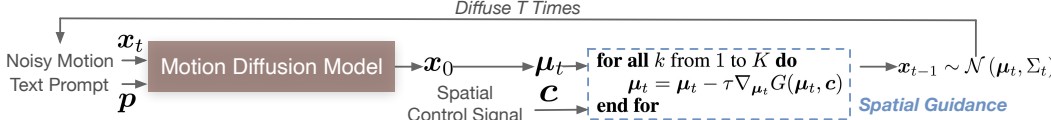

Figure 3: **Detailed illustration of our proposed spatial guidance**. The spatial guidance can effectively enforce the controlled joints to adhere to the input control signals.

yield unreasonable velocities or leg lengths, culminating along the generation process, which lead to unnatural generated motions. We still use the relative representations as input. To address the aforementioned limitation, we convert the relative representations to global ones in our proposed spatial guidance, allowing flexible control of any joints at any time, which will be introduced in detail in the next section.

## 3.2 Motion Generation with Flexible Spatial Control Signal

When the text prompt $p$ and spatial control signal $c$ are given together, how to ensure the generated motions adhere to both of them while remaining realistic is key to producing plausible motions. In this section, we will introduce our spatial and realism guidance to fulfill such an objective.

**Spatial guidance.** The architecture of spatial guidance is shown in Fig. 3. The core of our spatial guidance is an analytic function $G(\boldsymbol{\mu}_t, \boldsymbol{c})$ that assesses how closely the joint of the generated motion aligns with a desired spatial location $\boldsymbol{c}$. Following Dhariwal & Nichol (2021), the gradient of the analytic function is utilized to guide the generated motions in the desired direction. We employ the spatial guidance to perturb the predicted mean at every denoising step $t$[2]

$$\boldsymbol{\mu}_t = \boldsymbol{\mu}_t - \tau \nabla_{\boldsymbol{\mu}_t} G(\boldsymbol{\mu}_t, \boldsymbol{c}), \tag{2}$$

where $\tau$ controls the strength of the guidance. $G$ measures the L2 distance between the joint location of the generated motion and the spatial constraints:

$$G(\boldsymbol{\mu}, \boldsymbol{c}) = \frac{\sum_n \sum_j \sigma_{nj} \left\| \boldsymbol{c}_{nj} - \boldsymbol{\mu}_{nj}^g \right\|_2}{\sum_n \sum_j \sigma_{nj}}, \ \boldsymbol{\mu}^g = R(\boldsymbol{\mu}), \tag{3}$$

where $\sigma_{nj}$ is a binary value indicating whether the spatial control signal $\boldsymbol{c}$ contains a valid value at frame $n$ for joint $j$. $R(\cdot)$ converts the joint's local positions to global absolute locations. For simplicity, we omit the diffusion denoising step $t$ here. In this context, the global location of the pelvis at a specific frame can be determined through cumulative aggregation of rotations and translations from all the preceding frames. The locations of the other joints can also be ascertained through the aggregation of the pelvis position and the relative positions of the other joints.

Unlike existing approaches that convert global control signals to the relative locations w.r.t. the pelvis, which are non-existent or not accurate in-between the diffusion process, we propose to convert the generated motion to global coordinates. It eliminates ambiguity and thus empowers the model to incorporate flexible control signals over any joint at any time. Note that we still use the local human pose representations as the model's input and output. Consequently, the control signal is effective for all previous frames beyond the keyframe of the control signal as the gradients can be backpropagated to them, enabling the spatial guidance to densely perturb the motions even when the spatial constraints are extremely sparse. Moreover, as the positions of the remaining joints are relative to the pelvis position, spatial constraints applied to other joints can also influence the gradients on the pelvis position of previous frames. This property is desired. For instance, when one intends to reach for an object with a hand, adjustment of the pelvis position is usually needed, which would otherwise lead to unreasonable arm lengths in the generated motion. Note, however, that spatial control signals applied to the pelvis will not affect other joints. We address this problem using the realism guidance introduced below.

Our proposed spatial guidance is more effective than the classifier guidance used in other motion generation works (Rempe et al., 2023; Kulkarni et al., 2024; Karunratanakul et al., 2023) in terms of control accuracy. The key advantage lies in the fact that the gradient is calculated w.r.t the predicted mean $\boldsymbol{\mu}_t$, which only needs backpropagation through the lightweight function in Eq.(3). In contrast, previous works train a classifier or reward function and need gradient backpropagation through

---

[2]We should note that the denoising step $t$ should be distinguished from the frame number $n$.

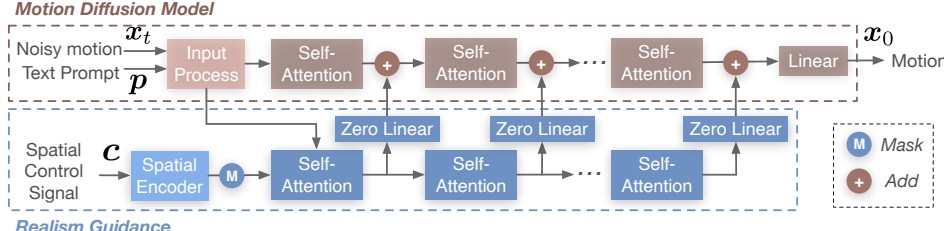

Figure 4: **Detailed illustration of our proposed realism guidance**. The realism guidance outputs the residuals w.r.t. the features in each attention layer of the motion diffusion model. These residuals can directly perturb the whole-body motion densely and implicitly.

a heavier model (*e.g.*, the entire motion diffusion model or a classifier), which is notably time-intensive. Thus, they guide the generated motion *only once* at each denoising diffusion step to maintain efficiency, which typically falls short of achieving the desired objective. Instead, we can afford to perturb the generated motion sequence for *multiple times*, largely improving the control accuracy. Specifically, we perturb $\mu_t$ by applying Eq.(2) iteratively for $K$ times at the denoising step t, which is set *dynamically* to balance the control accuracy and inference speed:

$$K = \begin{cases} K_e & \text{if } T_s \leq t \leq T, \\ K_l & \text{if } t \leq T_s. \end{cases} \tag{4}$$

We use $K_e = 10$, $K_l = 500$, and $T_s = 10$ in our experiments. In the early diffusion steps when $t \leq T_s$, the generated motion is of low quality. We enforce the spatial guidance for a small number of iterations. Later, as the quality of the motion improves when the diffusion step $t$ is large, intensive perturbations will be performed. The ablation study in Sec. 4.2 validates our design.

**Realism guidance.** Even though the spatial guidance can effectively enforce the controlled joints to adhere to the input control signals, it may leave other joints unchanged. For example, if we only control the pelvis position, the gradients of spatial guidance cannot be backpropagated to other joints due to the nature of the relative human pose representations and thus have no effect on other joints, as we mentioned earlier. It will lead to unrealistic motions. Moreover, since the perturbed position is only a small part of the whole motion, the motion diffusion model may ignore the change from the spatial guidance and fail to make appropriate modifications for the rest of the human joints, leading to incoherent human motion and foot sliding, as shown in Fig. 5 (b).

To address this issue, inspired by Zhang et al. (2023b), we propose realism guidance. Specifically, it is a trainable copy of the Transformer encoder in the motion diffusion model to learn to enforce the spatial constraints. The architecture of realism guidance is shown in Fig 4. The realism guidance takes in the same textual prompt $p$ as the motion diffusion model, as well as the spatial control signal $c$. Each of the Transformer layers in the original model and the new trainable copy are connected by a linear layer with both weight and bias initialized with zeros, so they have no effect of controlling at the beginning. As the training goes on, the realism guidance model learns the spatial constraints and adds the learned feature corrections to the corresponding layers in the motion diffusion model to amend the generated motions implicitly.

We use a spatial encoder $F$ to encode the spatial control signals $c$ at each frame independently, as shown in Fig. 4. To effectively handle the sparse control signals in time, we mask out the features at frames where there are no valid control signals, $\boldsymbol{f}_n = o_n F(\boldsymbol{c}_n)$. $o_n$ is a binary label that is an aggregation of $\sigma_{nj}$ in Eq.(3) such that $o_n$ is 1 (valid) if any of $\{\sigma_{nj}\}_{j=1}^J$ is 1. Otherwise, it is 0 (invalid). $\boldsymbol{f}_n$ are the features of spatial control signals at frame $n$, which are fed into the trainable copy of the Transformer. This helps the following attention layers know where the valid spatial control signals are and thus amend the corresponding features.

**Combination of spatial and realism guidance.** These two guidance are complementary in design, and both of them are necessary. The spatial guidance can change the position of corresponding control joints as well as the pelvis position to make the generated motion fulfill the spatial constraints. But it usually fails to amend the position of other joints that cannot receive the gradients, producing unreal and physically implausible motions. At the same time, although the realism guidance alone cannot ensure the generated motion tightly follows the spatial control signals, it amends the whole-body motion well, making up for the critical problem of spatial guidance. The combination of spatial guidance and realism guidance can effectively balance realistic human motion generation and the accuracy of incorporating spatial constraints. We ablate these two guidance in Sec. 4.2.

| Method | Joint | FID ↓ | R-precision ↑ (Top-3) | Diversity → | Foot skating ratio ↓ | Traj. err. ↓ (50 cm) | Loc. err. ↓ (50 cm) | Avg. err. ↓ |
|---|---|---|---|---|---|---|---|---|
| Real | - | 0.002 | 0.797 | 9.503 | 0.000 | 0.000 | 0.000 | 0.000 |
| MDM | | 0.698 | 0.602 | 9.197 | 0.1019 | 0.4022 | 0.3076 | 0.5959 |
| PriorMDM | Pelvis | 0.475 | 0.583 | 9.156 | 0.0897 | 0.3457 | 0.2132 | 0.4417 |
| GMD | | 0.576 | 0.665 | 9.206 | 0.1009 | 0.0931 | 0.0321 | 0.1439 |
| Ours (on pelvis) | | **0.218** | **0.687** | **9.422** | **0.0547** | **0.0387** | **0.0096** | **0.0338** |
| Ours (on all) | Pelvis | 0.322 | 0.691 | 9.545 | 0.0571 | 0.0404 | 0.0085 | 0.0367 |
| Ours (on all) | Left foot | 0.280 | 0.696 | 9.553 | 0.0692 | 0.0594 | 0.0094 | 0.0314 |
| Ours (on all) | Right foot | 0.319 | 0.701 | 9.481 | 0.0668 | 0.0666 | 0.0120 | 0.0334 |
| Ours (on all) | Head | 0.335 | 0.696 | 9.480 | 0.0556 | 0.0422 | 0.0079 | 0.0349 |
| Ours (on all) | Left wrist | 0.304 | 0.680 | 9.436 | 0.0562 | 0.0801 | 0.0134 | 0.0529 |
| Ours (on all) | Right wrist | 0.299 | 0.692 | 9.519 | 0.0601 | 0.0813 | 0.0127 | 0.0519 |
| Ours (on all) | Average | 0.310 | 0.693 | 9.502 | 0.0608 | 0.0617 | 0.0107 | 0.0404 |
| Ours (on all) | Cross | 0.624 | 0.672 | 9.016 | 0.0874 | 0.2147 | 0.0265 | 0.0766 |

Table 1: **Quantitative results on the HumanML3D test set.** *Ours (on pelvis)* means the model is only trained on pelvis control. *Ours (on all)* means the model is trained on all joints. *Joint (Average)* reports the average performance over all joints. *Joint (Cross)* reports the performance over the cross combination of joints. → means closer to real data is better.

| Method | Joint | FID ↓ | R-precision ↑ (Top-3) | Diversity → | Traj. err. ↓ (50 cm) | Loc. err. ↓ (50 cm) | Avg. err. ↓ |
|---|---|---|---|---|---|---|---|
| Real | - | 0.031 | 0.779 | 11.08 | 0.000 | 0.000 | 0.000 |
| PriorMDM | | 0.851 | **0.397** | 10.518 | 0.3310 | 0.1400 | 0.2305 |
| GMD | Pelvis | 1.565 | 0.382 | 9.664 | 0.5443 | 0.3003 | 0.4070 |
| Ours (on pelvis) | | **0.702** | **0.397** | **10.927** | **0.1105** | **0.0337** | **0.0759** |
| Ours (on all) | Average | 0.788 | 0.379 | 10.841 | 0.1433 | 0.0368 | 0.0854 |

Table 2: **Quantitative results on the KIT-ML test set.** *Ours (on pelvis)* means the model is only trained on pelvis control. *Ours (on all)* means the model is trained on all joints. *Joint (Average)* reports the average performance over all joints. → means closer to real data is better.

## 4 EXPERIMENTS

**Datasets.** We experiment on the popular HumanML3D (Guo et al., 2022a) dataset which contains 14,646 text-annotate human motion sequences from AMASS (Mahmood et al., 2019) and Human-Act12 (Guo et al., 2020) datasets. We also evaluate our method on the KIT-ML (Plappert et al., 2016) dataset with 3,911 sequences.

**Evaluation methods.** We adopt the evaluation protocol from Guo et al. (2022a). *Fréchet Inception Distance (FID)* reports the **naturalness** of the generated motion. *R-Precision* evaluates the **relevancy** of the generated motion to its text prompt, while *Diversity* measures the **variability** within the generated motion. In order to evaluate the controlling performance, following Karunratanakul et al. (2023), we report *foot skating ratio* as a proxy for the **incoherence** between trajectory and human motion and **physical plausibility**. We also report *Trajectory error*, *Location error*, and *Average error* of the locations of the controlled joints in the keyframes to measure the **control accuracy**. We provide detailed information of other metrics in the Appendix A.10. All the models are trained to generate 196 frames in our evaluation, where we use 5 sparsity levels in the controlling signal, including 1, 2, 5, 49 (25% density), and 196 keyframes (100% density). The time steps of keyframes are randomly sampled. We report the average performance over all density levels. In both training and evaluation, all models are provided with ground-truth trajectories as the spatial control signals.

### 4.1 COMPARISON TO OTHER METHODS

Since all previous methods, MDM (Tevet et al., 2023), PriorMDM (Shafir et al., 2024), and GMD (Karunratanakul et al., 2023) focus on controlling the pelvis only, we report the pelvis controlling performance for fair comparisons (*Joint: Pelvis*). All of these existing methods use the same pose representations and thus inherit the limitations detailed in 3.1. As a result, they only accept spatial constraints that are dense and over the pelvis alone. GMD changes the pelvis location from relative representation to absolute global ones so it can handle sparse control signals over the pelvis via a two-stage design. However, GMD only takes care of the pelvis location of the human body on the ground plane (xz positions). We retrain GMD to handle the full position of the pelvis (xyz position) to fairly compare with our method.

| Method | Joint | FID ↓ | R-precision ↑ (Top-3) | Diversity → 9.503 | Foot skating ratio ↓ | Traj. err. ↓ (50 cm) | Loc. err. ↓ (50 cm) | Avg. err. ↓ |
|---|---|---|---|---|---|---|---|---|
| Ours (on all) | | **0.310** | **0.693** | **9.502** | 0.0608 | **0.0617** | **0.0107** | **0.0385** |
| w/o spatial | Average | 0.351 | 0.691 | 9.506 | 0.0561 | 0.4285 | 0.2572 | 0.4137 |
| w/o realism | | 0.692 | 0.621 | 9.381 | 0.0909 | 0.2229 | 0.0606 | 0.1131 |
| Gradient w.r.t $x_t$ | | 0.336 | 0.691 | 9.461 | **0.0559** | 0.2590 | 0.1043 | 0.2380 |

Table 3: **Ablation studies** on the HumanML3D test set.

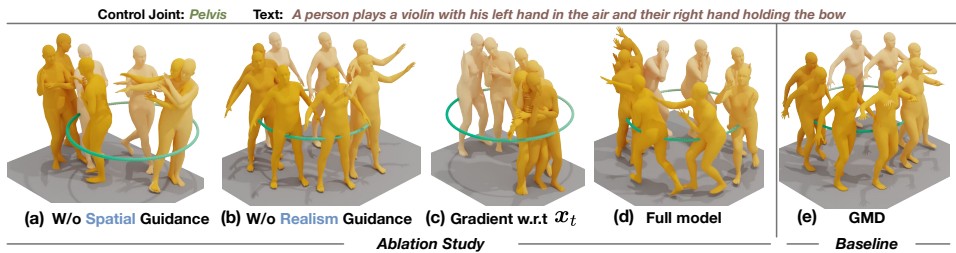

**Control Joint:** *Pelvis*  **Text:** *A person plays a violin with his left hand in the air and their right hand holding the bow*

(a) W/o **Spatial** Guidance  (b) W/o **Realism** Guidance  (c) Gradient w.r.t $x_t$  (d) **Full model**  (e) **GMD**

*Ablation Study* ————  *Baseline* ———

Figure 5: **Visual comparisons** of the ablation designs, our full model, and the baseline GMD.

The top part in Table 1 reports the comparisons of different methods on the HumanML3D dataset. Our method consistently outperforms all existing methods in the pelvis control over all metrics in terms of both realism and control accuracy. In particular, our approach has a significant reduction of $54.1\%$ in terms of *FID* compared to PriorMDM, proving that our proposed hybrid guidance generates much more realistic motions. Our method also surpasses the previous state-of-the-art method GMD by reducing *Avg. err.* of $79.2\%$. In addition, our foot skating ratio is the lowest compared to all other methods. We provide the complete table in the appendix.

More importantly, unlike previous approaches that can control the pelvis alone, our model can control over all joints using a single model. In the second part of Table 1, we report the performance in controlling each joint, where we consider *pelvis*, *left foot*, *right foot*, *head*, *left wrist*, and *right wrist*, given their common usage in the interactions with objects and the surrounding scene. We can see our model can achieve comparable performance in controlling the pelvis with only one model compared to the model specifically trained for controlling the pelvis only. The third part of Table 1 also show that the average controlling performance of each joint (*Joint: Average*) are comparable to the pelvis on both the HumanML3D and KIT-ML datasets. This largely simplifies the model training and usage, and provides more flexibility in controlling human motion generation. In the last row (*Joint: Cross*), we report the performance over the cross combination of joints. We randomly sample one possible combination for each sample during the evaluation. The results on KIT-ML dataset are reported in Table 2.

## 4.2 ABLATION STUDIES

We conduct several ablation experiments on HumanML3D to validate the effectiveness of our model's design choices. We summarize key findings below.

**Spatial guidance largely improves the controlling performance.** In Table 3, we compare our model (1st row) to a variant without any spatial guidance, *w/o spatial guidance* (2nd row) to show its effectiveness. The model with spatial guidance performs much better across all metrics of control accuracy (*Traj. err.*, *Loc. err.*, and *Avg. err.*) and shows $90\%$ decrease in *Avg. err.*. Fig. 5(a) validates this observation, where we can see the generated motion cannot tightly follow the spatial constraints without spatial guidance. These results show that the spatial guidance is effective.

**Computing gradients w.r.t $\mu_t$ is effective.** In spatial guidance, we calculate the gradient w.r.t the predicted $\mu_t$. To show the effectiveness of this design, we report the performance of a variant which computes the gradient w.r.t the input noised motion $x_t$, in Table 3 *Gradient w.r.t $x_t$* (last row). Following Karunratanakul et al. (2023), we only perturb the controlled joints once at each diffusion step, partially due to the long-running time of 99s. In our spatial guidance, it takes 121s to perturb the joints multiple times. Our spatial guidance produces $83.8\%$ lower *Avg. err.*, validating that our design is much more effective compared to the similar operations used in Karunratanakul et al. (2023). Fig. 5 (c) validates this observation.

**Realism guidance is critical for generating coherent motions.** As shown in Table 3, compared to a variant without realism guidance, *w/o realism guidance* (3rd row), our proposed model leads to $50\%$

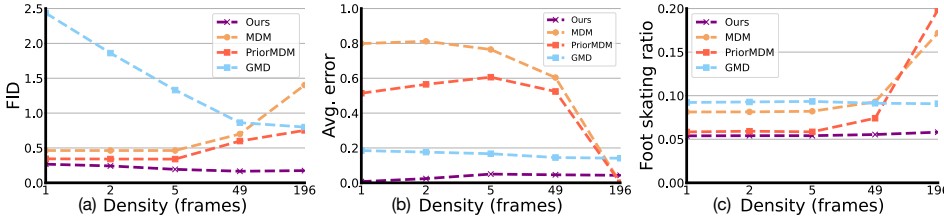

Figure 6: **Balancing inference time and Avg. Error** by varying $T_s$, $K_e$, and $K_l$ in spatial guidance. The performance is reported on pelvis control on the HumanML3D with dense control signals.

Figure 7: **Varying the density of spatial signal**. The performance is reported on pelvis control on the HumanML3D dataset with the x-axis in logarithmic scale. All metrics are the lower the better.

decrease in *FID*. Fig. 5(b) visualizes the generated motions when removing the realism guidance. In this case, the model cannot amend the rest of the joints by correctly fusing the information in both the input textual prompt and spatial control signals, yielding unreal and incoherent motions.

### 4.3 DEEPER DIVE INTO OMNICONTROL

**Balancing the inference time and Average Error.** The spatial guidance in OmniControl adopts an iterative strategy to perturb the predicted mean $\mu_t$ at each diffusion step. We explore the effect of varying the dynamic number iterations ($K_e$ and $K_l$ in Eq.(4)) in Fig. 6. We see that more iterations in the early stage of the diffusion process do not necessarily lead to better performance. So we use $K_e << K_l$. We vary $T_s$ in Fig. 6 (a). When setting $T_s$ smaller than 10, the inference time slightly drops but the Average Error increases. On the contrary, a large $T_s$ causes a much larger inference time is much larger (121s *vs* 143s). So we set $T_s = 10$ for a trade-off. We vary $K_e$ in Fig. 6 (b), in which large $K_e$ ($> 10$) reports steady performance but much more time in inference. We vary $K_l$ in Fig. 6 (c), where $K_l = 500$ is an appropriate setting to balance inference time and Average Error.

**Varying the density of the spatial signal.** We report the performance of different models in different density levels in Fig. 7. Under all density levels, GMD's performance is consistently worse than ours. Regarding MDM and PriorMDM, their *FID* and *Foot skating ratio* metrics significantly increase as the density increases while ours remain stable. When the spatial control signal is dense, the *Avg. error* of MDM and PriorMDM are 0 because of the properties of the inpainting method. However, substantially high *FID* and *Foot skating ratio* indicate that both of them cannot generate realistic motions and fail to ensure the coherence between the controlling joints and the rest, resulting in physically implausible motions. It can be clearly seen that our method is significantly more robust to different density levels than existing approaches.

**Controlling multiple joints together enables downstream applications.** We demonstrate the OmniControl can employ a single model to support controlling multiple joints together. This new capability enables a set of downstream applications, as shown in Fig. 1 (last column), which supports correctly connecting isolated human motion with the surrounding objects and scenes.

### 5 CONCLUSION

We presented OmniControl, an effective method that controls any joint of humans at any time for text-based human motion generation. OmniControl works by combining the spatial and realism guidance that are highly complementary, enabling realistic human motion generation while conforming to the input spatial control signals. Extensive experimental results and ablation studies on HumanML3D and KIT-ML datasets are provided to validate the effectiveness of OmniControl.

**Acknowledgement.** YX and HJ would like to acknowledge the support from Google Cloud Program Credits and a gift fund from Google.

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

# A APPENDIX

## A.1 PSEUDO CODE

---

**Algorithm 1 OmniControl**'s inference

---

**Require:** A motion diffusion model $M$, a realism guidance model $S$, spatial control signals $c$ (if any), text prompts $p$ (if any).

1: $x_T \sim \mathcal{N}(0, I)$
2: **for all** $t$ from $T$ to 1 **do**
3:     $\{f\} \leftarrow S(x_t, t, p, c; \phi)$          # **Realism guidance model**
4:     $x_0 \leftarrow M(x_t, t, p, \{f\}; \theta)$          # **Model diffusion model**
5:     $\mu_t, \Sigma_t \leftarrow \mu(x_0, x_t), \Sigma_t$
6:     **for all** $k$ from 1 to $K$ **do**          # **Spatial guidance**
7:         $\mu_t = \mu_t - \tau \nabla_{\mu_t} G(\mu_t, c)$
8:     **end for**
9:     $x_{t-1} \sim \mathcal{N}(\mu_t, \Sigma_t)$
10: **end for**
11: **return** $x_0$

---

| Methods | Ours | MDM | PriorMDM | GMD |
|---|---|---|---|---|
| Time (hours) | 29.0 | 72.0 | 11.5 | 39.04 |

Table 4: **Training time.** The training time of the baseline methods is from their paper. Ours is trained on a single NVIDIA RTX A5000 GPU. GMD Karunratanakul et al. (2023) is trained on a single NVIDIA RTX 3090. MDM Tevet et al. (2023) and PriorMDM Shafir et al. (2024) are trained on a single NVIDIA GeForce RTX 2080 Ti GPU.

## A.2 MORE IMPLEMENTATION DETAILS

**Training details.** We implemented our model using Pytorch with training on 1 NVIDIA A5000 GPU. Batch size $b = 64$. We use AdamW optimizer (Loshchilov & Hutter, 2017), and the learning rate is $1e - 5$. It takes 29 hours to train on a single A5000 GPU with 250,000 iterations in total. We compare our method to the baselines in terms of training time in Table 4.

**Model details.** Our baseline motion diffusion model is based on MDM (Tevet et al., 2023). Both the motion diffusion model and realism guidance model resume the pretrain weights from Tevet et al. (2023) and are fine-tuned together. Similar to MDM, we use the CLIP (Radford et al., 2021) model to encode text prompts and the generation process is in a classifier-free (Ho & Salimans, 2021) manner. The input process in Fig. 4 mainly consists of a CLIP-based (Radford et al., 2021) textual embedding to encode the text prompt and linear layers to encode the noisy motion. Then, the encoded text and encoded noisy motion will be concatenated as the input to the self-attention layers. The spatial encoder in Fig. 4 consists of four linear layers to encode the spatial control signals. The dimension of encoded spatial control signals $f_n$ is (L, B, C), where $L = 196$ is the sequence length, $B$ is the batch size, and $C = 512$ is the feature dimension. The spatial guidance is also used in training time. In the training stage, the prompt $p$ is randomly masked for classifier-free learning (Ho & Salimans, 2021). We utilize DDPM (Ho et al., 2020) with $T = 1000$ denoising steps. The control strength $\tau = \frac{20\hat{\Sigma}_t}{V}$, where $V$ is the number of frames we want to control (density) and $\hat{\Sigma}_t = \min(\Sigma_t, 0.01)$.

**Experiment details.** Tevet et al. (2023); Shafir et al. (2024) naturally cannot handle the sparse control signals due to the relative pelvis representation detailed in 3.1 in the main paper. To conduct these two methods with sparse control signals, we insert the ground truth velocity at specific times.

### A.3 Inference time

We report the inference time of our submodules, our full pipeline, and baseline methods in Table 5. The inference time is measured on an NVIDIA A5000 GPU. When comparing to GMD, we use the inference time reported in its paper.

| Sub-Modules | Realism Guidance | MDM | Spatial G. $K = K_e$ | Spatial G. $K = K_l$ | Methods *Overall* | Ours | MDM | GMD |
|---|---|---|---|---|---|---|---|---|
| Time (ms) | 19.3 | 18.3 | 42.5 | 1531.0 | Time (s) | 121.5 | 39.2 | 110.0 |

Table 5: **Inference time.** We report the time for baselines and each submodule of ours. The MDM in Sub-modules means the motion generation model we use in each diffusion step. The MDM in Methods *Overall* is Tevet et al. (2023).

### A.4 More Discussions about Controlling multiple joints

We report the quantitative results of controlling multiple joints in the last row of Table 1. There are 57 possible combinations for six types of joints. Since running an evaluation for each of them is costly, it's impractical to evaluate all the combinations. Instead, we randomly sample one possible combination for each motion sequence for evaluation. The performance is lower compared to the single-joint control (not an apple-to-apple comparison as the ground-truths are different). Nevertheless, the results show that controlling multiple joints is harder than a single one due to the increased degrees of freedom.

### A.5 Limitations and future plan

The problem with our approach is that there are still a lot of foot skating cases. The realism guidance is not a perfect module when used to amend the whole-body motion from the input spatial control signals. We are interested in exploring more effective designs to improve the realism and physical plausibility of human motion generation. In addition, some physical constraints (Yuan et al., 2023) can be used to reduce the foot skating ratio.

Another significant limitation of the diffusion approach arises from its extended inference time, necessitating roughly 1,000 forward passes to produce a single result. As diffusion models persist in their development (Lu et al., 2022; Lyu et al., 2022; Salimans & Ho, 2022), we are inclined to explore, in future work, strategies to expedite computational speed.

Although OmniControl can be used to control multiple joints without other special designs or fine-tuning, in some cases when the spatial control signals for two joints are conflicted, our method usually produces unnatural motions. We will explore more to improve this problem in future work.

### A.6 Why didn't we use global pose representation for human motion generation

The human pose representation suggested by Guo et al. (2022a) is easier to learn and produce realistic human motions because it leverages the human skeleton prior. However, this representation is not friendly for inpainting-based methods, detailed in 3.1. One question is whether we can use the global representation for all joints of humans. We try to train the MDM (Tevet et al., 2023) using the global representation, in which the human pose is represented with the global position of joints. In this case, $D = 66$ (22 joints) on HumanML3D dataset or $D = 63$ (21 joints) on KIT-ML dataset. We found the model cannot converge, and produce unreasonable human poses, as shown in Fig. 8.

### A.7 Why didn't we use the global pose representation proposed in Liang et al. (2024)

Liang et al. (2024) propose a non-canonicalization representation for multi-person interaction motion. Instead of transforming joint positions and velocities to the root frame, they keep them in the

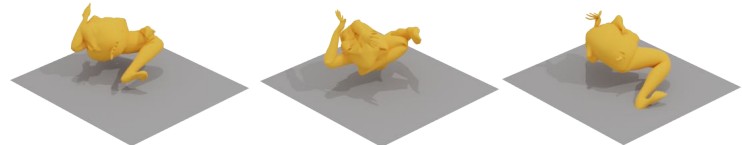

Figure 8: With global pose representation, the model cannot produce reasonable human poses on the HumanML3D dataset.

world frame. To avoid the unreasonable human poses shown in A.6, they introduce bone length loss to constrain the global joint positions to satisfy skeleton consistency, which implicitly encodes the human body's kinematic structure. We train MDM (Tevet et al., 2023) with this global representation and bone length loss, and report the performance in Tab. 6. There is a significant drop in performance when converted to global representation for text-based human motion generation. Therefore, global coordinates are not an optimal choice for our task.

| Representation | FID ↓ | R-precision ↑ (Top-3) | Diversity → 9.503 |
|---|---|---|---|
| Relative | 0.544 | 0.611 | 9.559 |
| Global | 1.396 | 0.585 | 9.195 |

Table 6: **Text-to-motion evaluation on the HumanML3D (Guo et al., 2022a) dataset.** Comparison between relative presentation (Guo et al., 2022a) and global representation (Liang et al., 2024).

### A.8 WHY DIDN'T WE REPORT THE FOOT SKATING RATIO ON KIT-ML DATASET

The data quality of KIT-ML is relatively low. The foot height on KIT-ML is not necessarily close to 0, even when the foot is on the ground, and there are a lot of foot skating cases in the ground truth motions. We, therefore do not evaluate the skating ratio on KIT-ML because it cannot be evaluated accurately.

### A.9 MORE DISCUSSION ABOUT FIG. 7

**In Fig. 7 (a), the higher density leads to lower FID and Foot skating ratio for ours and GMD, while it's the opposite for the other two methods (MDM and PriorMDM).** Ideally, higher density should lead to better performance if the generated motion could accurately follow the control signal, and make corresponding adjustments to the other joints to make the motion realistic and natural. This is not the case for MDM and PriorMDM since they cannot effectively modify whole-body motion according to the input control signal. When the density is higher (the constraint is stricter) and other joints are NOT adjusted effectively to compensate for the more rigidity in the control signal, they will produce unnatural results, thus leading to higher FID and higher foot skating ratio. On the contrary, GMD and ours, which are specially designed for both text and spatial control signal conditions, can efficiently adjust the whole-body motion and better leverage the context information in the input signal, yielding better performance when the density is higher.

**In Fig. 7 (b), the Avg. error of MDM and PriorMDM is zero when using dense control signals.** The Avg. error of MDM and PriorMDM is zero due to the inpainting property. Inpainting-based methods aim to reconstruct the rest of joint motions based on the given control signals over one or more control joints. The input control signals won't be changed during this process, i.e., the output motion over the control joints remains the same as the input control signal. As a result, the Avg. error is zero.

### A.10 DETAILED INFORMATION OF THE METRICS

*Fréchet Inception Distance (FID)* reports the **naturalness** of the generated motion. *R-Precision* evaluates the **relevancy** of the generated motion to its text prompt, while *Diversity* measures the

**variability** within the generated motion. In order to evaluate the controlling performance, following Karunratanakul et al. (2023), we report *foot skating ratio* as a proxy for the **incoherence** between trajectory and human motion and **physical plausibility**. It measures the proportion of frames in which either foot skids more than a certain distance (2.5 cm) while maintaining contact with the ground (foot height < 5 cm). We also report *Trajectory error*, *Location error*, and *Average error* of the locations of the controlled joints in the keyframes to measure the **control accuracy**. Trajectory error is the ratio of unsuccessful trajectories, defined as those with any keyframe location error exceeding a threshold. Location error is the ratio of keyframe locations that are not reached within a threshold distance. Average error measures the mean distance between the generated motion locations and the keyframe locations measured at the keyframe motion steps.

## A.11  The source of the spatial control signal

In both training and evaluation, all models are provided with ground-truth trajectories as the spatial control signals. In the visualizations or video demos, the spatial control signals are manually designed. In the downstream applications, we envision there may be two ways to collect the spatial guidance trajectories. First, for practical applications in industries such as 3D animation, the user (designer) may provide such guidance trajectories as part of the application development. Second, the spatial guidance may be from the interactions and constraints of the scene/object. For example, the height of the ceiling or the position of the chair (and thus the position of the controlled joint), as we show in the last column of Fig. 1 in the paper. In both cases, the spatial guidance trajectories can be efficiently collected.

## A.12  The source of object models used in the last column of Fig. 1

All these object models are licensed under Creative Commons Attribution 4.0 International License. Specifically, Chair: external link; Ceiling fan: external link. Handrail: external link.

## A.13  All evaluation results

In Table 7 and Table 8, we first present the detailed performance of OmniControl across five sparsity levels, which is trained for pelvis control on the HumanML3D and KIT-ML test set. Subsequently, in Table 10, and Table 9, we showcase the comprehensive results of OmniControl in controlling various joints (pelvis, left foot, right foot, head, left wrist, and right wrist). BUN in Table 9 means body upper neck. As shown in table 10, the Traj. err. and Loc. err. are sometimes zeros when the density is low since the definitions of these two metrics are not strict. As discussed in A.10, Traj. err. (50 cm) is the ratio of unsuccessful trajectories. The unsuccessful trajectories are defined as the trajectories with any keyframe whose location error exceeds a threshold (50 cm). And Loc. err. (50 cm) is the ratio of unsuccessful keyframes whose location error exceeds a threshold (50 cm). When the density is low (e.g., only have spatial control signal in one keyframe), it is easier for all samples to meet this threshold and thus achieve zero errors.

| Keyframe | FID ↓ | R-precision ↑ (Top-3) | Diversity → 9.503 | Foot skating ratio ↓ | Traj. err. ↓ (50 cm) | Loc. err. ↓ (50 cm) | Avg. err. ↓ |
|---|---|---|---|---|---|---|---|
| 1 | 0.272 | 0.675 | 9.547 | 0.0533 | 0.0000 | 0.0000 | 0.0079 |
| 2 | 0.251 | 0.683 | 9.498 | 0.0540 | 0.0195 | 0.0103 | 0.0253 |
| 5 | 0.210 | 0.691 | 9.345 | 0.0541 | 0.0645 | 0.0203 | 0.0507 |
| 49 (25%) | 0.179 | 0.689 | 9.410 | 0.0558 | 0.0635 | 0.0108 | 0.0452 |
| 196 (100%) | 0.181 | 0.698 | 9.311 | 0.0564 | 0.0459 | 0.0066 | 0.0398 |

Table 7: **The detailed results on the HumanML3D test set.** OmniControl is only trained for pelvis control on the HumanML3D test set.

| Keyframe | FID ↓ | R-precision ↑ (Top-3) | Diversity → 11.08 | Traj. err. ↓ (50 cm) | Loc. err. ↓ (50 cm) | Avg. err. ↓ |
|---|---|---|---|---|---|---|
| 1 | 0.723 | 0.406 | 10.922 | 0.0099 | 0.0099 | 0.0253 |
| 2 | 0.774 | 0.391 | 10.980 | 0.0355 | 0.0213 | 0.0496 |
| 5 | 0.731 | 0.390 | 10.906 | 0.1562 | 0.0494 | 0.0969 |
| 49 (25%) | 0.700 | 0.403 | 10.920 | 0.2003 | 0.0504 | 0.1108 |
| 196 (100%) | 0.583 | 0.394 | 10.910 | 0.1506 | 0.0375 | 0.0968 |

Table 8: **The detailed results on the KIT-ML test set.** OmniControl is only trained for pelvis control on the KIT-ML test set.

| Number of Keyframe | Joint | FID ↓ | R-precision ↑ (Top-3) | Diversity → 11.08 | Traj. err. ↓ (50 cm) | Loc. err. ↓ (50 cm) | Avg. err. ↓ |
|---|---|---|---|---|---|---|---|
| 1 | Pelvis | 1.077 | 0.364 | 10.794 | 0.0099 | 0.0099 | 0.0265 |
| 2 | Pelvis | 1.152 | 0.361 | 10.725 | 0.0256 | 0.0156 | 0.0426 |
| 5 | Pelvis | 1.235 | 0.361 | 10.789 | 0.1477 | 0.0469 | 0.0920 |
| 49 (25%) | Pelvis | 1.306 | 0.365 | 10.789 | 0.2301 | 0.0570 | 0.1293 |
| 196 (100%) | Pelvis | 1.417 | 0.374 | 10.823 | 0.2443 | 0.0625 | 0.1470 |
| 1 | BUN | 1.056 | 0.365 | 10.804 | 0.0043 | 0.0043 | 0.0188 |
| 2 | BUN | 1.170 | 0.372 | 10.742 | 0.0156 | 0.0107 | 0.0261 |
| 5 | BUN | 1.262 | 0.361 | 10.777 | 0.1037 | 0.0349 | 0.0662 |
| 49 (25%) | BUN | 1.285 | 0.371 | 10.782 | 0.2230 | 0.0544 | 0.1171 |
| 196 (100%) | BUN | 0.866 | 0.395 | 10.981 | 0.2216 | 0.0547 | 0.1342 |
| 1 | Left foot | 0.601 | 0.384 | 10.824 | 0.0014 | 0.0014 | 0.0184 |
| 2 | Left foot | 0.567 | 0.378 | 10.818 | 0.0170 | 0.0107 | 0.0277 |
| 5 | Left foot | 0.605 | 0.388 | 10.811 | 0.1420 | 0.0435 | 0.0770 |
| 49 (25%) | Left foot | 0.589 | 0.381 | 10.850 | 0.3210 | 0.0737 | 0.1418 |
| 196 (100%) | Left foot | 0.537 | 0.389 | 10.972 | 0.3665 | 0.0788 | 0.1670 |
| 1 | Right foot | 0.596 | 0.382 | 10.843 | 0.0014 | 0.0014 | 0.0163 |
| 2 | Right foot | 0.575 | 0.385 | 10.844 | 0.0156 | 0.0092 | 0.0260 |
| 5 | Right foot | 0.601 | 0.381 | 10.886 | 0.1108 | 0.0366 | 0.0772 |
| 49 (25%) | Right foot | 0.615 | 0.378 | 10.871 | 0.3253 | 0.0699 | 0.1440 |
| 196 (100%) | Right foot | 0.575 | 0.376 | 10.893 | 0.3764 | 0.0774 | 0.1660 |
| 1 | Left wrist | 0.545 | 0.386 | 10.896 | 0.0014 | 0.0014 | 0.0172 |
| 2 | Left wrist | 0.567 | 0.395 | 10.884 | 0.0114 | 0.0071 | 0.0231 |
| 5 | Left wrist | 0.601 | 0.392 | 10.820 | 0.1065 | 0.0344 | 0.0751 |
| 49 (25%) | Left wrist | 0.632 | 0.391 | 10.891 | 0.2699 | 0.0627 | 0.1531 |
| 196 (100%) | Left wrist | 0.569 | 0.389 | 10.890 | 0.3139 | 0.0694 | 0.1824 |
| 1 | Right wrist | 0.598 | 0.386 | 10.834 | 0.0014 | 0.0014 | 0.0163 |
| 2 | Right wrist | 0.574 | 0.379 | 10.833 | 0.0185 | 0.0114 | 0.0249 |
| 5 | Right wrist | 0.636 | 0.379 | 10.817 | 0.0944 | 0.0341 | 0.0742 |
| 49 (25%) | Right wrist | 0.594 | 0.374 | 10.921 | 0.2656 | 0.0615 | 0.1516 |
| 196 (100%) | Right wrist | 0.631 | 0.392 | 10.836 | 0.3068 | 0.0665 | 0.1819 |

Table 9: **The detailed results of OmniControl on the KIT-ML test set.**

| Number of Keyframe | Joint | FID ↓ | R-precision ↑ (Top-3) | Diversity → 9.503 | Foot skating ratio ↓ | Traj. err. ↓ (50 cm) | Loc. err. ↓ (50 cm) | Avg. err. ↓ |
|---|---|---|---|---|---|---|---|---|
| 1 | Pelvis | 0.333 | 0.672 | 9.517 | 0.0616 | 0.0010 | 0.0010 | 0.0064 |
| 2 | Pelvis | 0.365 | 0.681 | 9.630 | 0.0589 | 0.0137 | 0.0068 | 0.0203 |
| 5 | Pelvis | 0.278 | 0.705 | 9.582 | 0.0575 | 0.0537 | 0.0154 | 0.0434 |
| 49 (25%) | Pelvis | 0.297 | 0.700 | 9.533 | 0.0552 | 0.0732 | 0.0108 | 0.0567 |
| 196 (100%) | Pelvis | 0.338 | 0.699 | 9.462 | 0.0525 | 0.0605 | 0.0087 | 0.0567 |
| 1 | Head | 0.395 | 0.683 | 9.469 | 0.0565 | 0.0000 | 0.0000 | 0.0073 |
| 2 | Head | 0.416 | 0.686 | 9.456 | 0.0574 | 0.0049 | 0.0024 | 0.0093 |
| 5 | Head | 0.309 | 0.700 | 9.467 | 0.0602 | 0.0361 | 0.0113 | 0.0248 |
| 49 (25%) | Head | 0.267 | 0.705 | 9.554 | 0.0541 | 0.0869 | 0.0138 | 0.0605 |
| 196 (100%) | Head | 0.290 | 0.704 | 9.453 | 0.0500 | 0.0830 | 0.0119 | 0.0726 |
| 1 | Left foot | 0.332 | 0.690 | 9.548 | 0.0676 | 0.0000 | 0.0000 | 0.0071 |
| 2 | Left foot | 0.346 | 0.690 | 9.536 | 0.0681 | 0.0059 | 0.0029 | 0.0097 |
| 5 | Left foot | 0.266 | 0.700 | 9.547 | 0.0741 | 0.0420 | 0.0119 | 0.0275 |
| 49 (25%) | Left foot | 0.240 | 0.698 | 9.634 | 0.0794 | 0.1357 | 0.0172 | 0.0536 |
| 196 (100%) | Left foot | 0.218 | 0.702 | 9.500 | 0.0570 | 0.1133 | 0.0151 | 0.0590 |
| 1 | Right foot | 0.368 | 0.677 | 9.419 | 0.0621 | 0.0000 | 0.0000 | 0.0077 |
| 2 | Right foot | 0.421 | 0.692 | 9.453 | 0.0621 | 0.0068 | 0.0039 | 0.0103 |
| 5 | Right foot | 0.318 | 0.695 | 9.414 | 0.0739 | 0.0527 | 0.0160 | 0.0320 |
| 49 (25%) | Right foot | 0.254 | 0.721 | 9.565 | 0.0801 | 0.1475 | 0.0227 | 0.0604 |
| 196 (100%) | Right foot | 0.234 | 0.719 | 9.561 | 0.0556 | 0.1260 | 0.0173 | 0.0617 |
| 1 | Left wrist | 0.371 | 0.676 | 9.400 | 0.0579 | 0.0000 | 0.0000 | 0.0085 |
| 2 | Left wrist | 0.404 | 0.672 | 9.544 | 0.0561 | 0.0039 | 0.0020 | 0.0106 |
| 5 | Left wrist | 0.322 | 0.687 | 9.468 | 0.0597 | 0.0312 | 0.0104 | 0.0341 |
| 49 (25%) | Left wrist | 0.249 | 0.684 | 9.440 | 0.0561 | 0.1855 | 0.0289 | 0.1002 |
| 196 (100%) | Left wrist | 0.172 | 0.682 | 9.329 | 0.0515 | 0.1797 | 0.0256 | 0.1109 |
| 1 | Right wrist | 0.388 | 0.682 | 9.537 | 0.0611 | 0.0000 | 0.0000 | 0.0078 |
| 2 | Right wrist | 0.374 | 0.690 | 9.527 | 0.0622 | 0.0059 | 0.0029 | 0.0112 |
| 5 | Right wrist | 0.276 | 0.688 | 9.610 | 0.0616 | 0.0479 | 0.0125 | 0.0348 |
| 49 (25%) | Right wrist | 0.246 | 0.700 | 9.465 | 0.0599 | 0.1855 | 0.0259 | 0.0978 |
| 196 (100%) | Right wrist | 0.211 | 0.700 | 9.456 | 0.0555 | 0.1670 | 0.0222 | 0.1077 |

Table 10: **The detailed results of OmniControl on the HumanML3D test set.**

