# OpenReview forum: "OmniControl: Control Any Joint at Any Time for Human Motion Generation"
_ICLR.cc/2024/Conference — ICLR 2024 poster_

### Official Review · Reviewer_4D6Y · 2023-10-26

**Soundness:** 3 good
**Presentation:** 3 good
**Contribution:** 2 fair
**Rating:** 6
**Confidence:** 4

**Summary:**

This paper introduces a generative model for text-conditional human motion synthesis offering detailed control over joint positions in each frame. The authors propose two guidance mechanisms, spatial and realism guidance, that aim to generate human motion which closely adheres to the guidance while maintaining realism. The effectiveness of these designs is established through experiments, supported by comprehensive data and high-quality visualizations.

**Strengths:**

1. The paper considers the novel task of guiding individual joints spatially in human motion generation. Experiments conducted on the HumanML3D and KIT-ML datasets show promising results.

2. The experimental setup is robust and the accompanying visualizations are of high quality, reflecting the authors' meticulous efforts in this research.

**Weaknesses:**

1. Previous works guiding the position of the pelvis could potentially be intuitively adapted to joint positions with proper coordinate transformations, which undermines the novelty of this work, making it seem more of an incremental step rather than a solution to fundamental problems in the field.

2. Despite the novelty of controlling all joints at any frame, the paper lacks a discussion on efficient collection of the spatial guidance trajectory, particularly considering the fact that the joint positions are no more relative to the pelvis. This aspect is crucial for practical applications in industries such as 3D animation. Moreover, the paper does not discuss the model's tolerance for inherently unnatural guidance (*e.g.*, manually drawn trajectories or those from different datasets).

3. There are some formatting errors (*e.g.*, the misuse of `\citep` instead of `\citet` in some citations).

**Questions:**

In most scenarios, the text input and the spatial guidance seem redundant. Can the model effectively comprehend and follow instructions that appear only in one modality? (*e.g.*, providing spatial guidance only on the foot or pelvis, while the hand activity is only described via the text input?)

---

> ### Author Response · Authors · 2023-11-21
> **Response to Reviewer 4D6Y (1/2)**
>
> Thank you for the constructive and helpful feedback. We respond below to your questions and comments:
>
> ```
> Q1: Previous works guiding the position of the pelvis could potentially be intuitively adapted to joint positions with proper coordinate transformations…
> ```
> **A1**: Do you mean that, if we know the coordinate transformation between the pelvis and other joints, we may be able to control them together using previous methods?
>
> In general, we know in advance neither the position of the pelvis nor the coordinate transformations between the pelvis and other joints that we’d to control, as discussed in the introduction (paragraph 2) and Section 3.1 (human pose representations). Thus adapting the previous pelvis-control-only methods to other joints is not feasible.
>
> If it’s not what you mean, we would greatly appreciate it if you can clarify the question.
>
>
> &nbsp;
>
> ```
> Q2: The paper lacks a discussion on efficient collection of the spatial guidance trajector…
> ```
> **A2**: We envision there may be two ways to collect the spatial guidance trajectories. First, for practical applications in industries such as 3D animation, the user (designer) may provide such guidance trajectories as part of the application development. Second, the spatial guidance may be from the interactions and constraints of the scene/object. For example, the height of the ceiling or the position of the chair (and thus the position of the controlled joint), as we show in the last column of [Figure 1](https://github.com/OmniControl24/OmniControl24.github.io/blob/main/static/images/figure1.jpg) in the paper. In both cases, the spatial guidance trajectories can be efficiently collected.
> The discussion has been added to **Appendix A.11** of the revised paper.
>
> &nbsp;
>
> ```
> Q3: The paper does not discuss the model's tolerance for inherently unnatural guidance (e.g., manually drawn trajectories…
> ```
> **A3**: In all the figures and supplementary video, the input control signals are manually drawn (e.g. the circle, straight, and sine lines in [Figure 1](https://github.com/OmniControl24/OmniControl24.github.io/blob/main/static/images/figure1.jpg) and [supplementary video](https://youtu.be/29M-Shs1Orw?t=27)). To better resolve your concern, we added some results with extremely unnatural spatial control signals in the updated supplementary video [(from 4:41 to 5:23)](https://youtu.be/29M-Shs1Orw?t=284) and penultimate section of the [website](https://omnicontrol24.github.io/). These results show that our model has a good tolerance for unnatural trajectories (teleportation, spiral forward).
>
> &nbsp;
>
> ```
> Q4: Misuse of \citep instead of \citet in some citations…
> ```
> **A4**: Thank you for pointing this out. We have revised these misuses in the citations.
>
> &nbsp;
>
> ```
> Q5: The text input and the spatial guidance seem redundant…
> ```
> **A5**: In some cases especially when the human is interacting with the scene or object, the spatial control signals are usually provided as global locations of joints of interest in keyframes **as they are hard to convey in the textual prompt**. For instance, in the low-ceiling scenario shown in the last column of [Figure 1](https://github.com/OmniControl24/OmniControl24.github.io/blob/main/static/images/figure1.jpg) in the paper, it is infeasible to specify the spatial constraints in the textual prompt precisely. Even if we do, there is no guarantee that the model will faithfully incorporate them into the human motion generation, which will lead to unsatisfactory user experiences. Therefore, it is more natural and intuitive to provide spatial control signals as a separate input. At the same time, the textual description is helpful for conveying high-level semantic guidance, such as “playing the violin”. Therefore, the text prompt and spatial guidance are complementary to each other instead of redundant.

---

> ### Author Response · Authors · 2023-11-21
> **Response to Reviewer 4D6Y (2/2)**
>
> ```
> Q6: Can the model effectively comprehend and follow instructions that appear only in one modality? (e.g., providing spatial guidance only on the foot or pelvis, while the hand activity is only described via the text input?)
> ```
> **A6**: Yes it can. In the example in the top left of [Figure 1](https://github.com/OmniControl24/OmniControl24.github.io/blob/main/static/images/figure1.jpg), the spatial control signal is applied to the pelvis (walk along the circle), while the text is used to describe the hand activity (play the violin).
> As can be seen, our model can effectively comprehend and follow instructions on separate joints (pelvis and hand). We provide the results with only a text prompt or only a trajectory as the condition in the supplementary video [(From 5:23 to 5:45)](https://youtu.be/29M-Shs1Orw?t=325) and the last section of the anonymous [website](https://omnicontrol24.github.io/). The results show that our model can follow the instructions that appear only in one modality.
>
> We’d like to note that since the text input and the control signals describe the motion of different joints, they are complementary instead of redundant.

---

> > ### Author Response · Authors · 2023-11-22
> > **Follow up with Reviewer 4D6Y**
> >
> > Dear Reviewer 4D6Y,
> >
> > Thank you for your detailed review and the valuable feedback! We would love to hear your thoughts about our rebuttal, including whether it sufficiently addresses your concerns and questions. Any feedback is welcome and greatly appreciated!
> >
> > Sincerely,
> > Authors of OmniControl

---

### Official Review · Reviewer_gCs4 · 2023-10-29

**Soundness:** 2 fair
**Presentation:** 2 fair
**Contribution:** 3 good
**Rating:** 6
**Confidence:** 5

**Summary:**

The authors propose a human motion generation method with the ability of manipulating any joint at any time. The method can take the language prompt as condition and both the spatial guidances and realism guidance as constraints. Compared to previous method, including MDM, based on which the method is developed, the proposed method showcases more flexibility for downstream applications. With a single model, the proposed OmniControl sets a new SOTA to control both the pelvis and other joints in motion generation.

**Strengths:**

- The method allows using spatial guidance to constraint the generated motion sequence. The constraint can be put to any joint instead of just pelvis.
- By combining the realism guidance, the conditionally generated motion sequences can be expected to be more natural under the spatial constraint. The realism guidance is a trainable copy of encoder to enforce the spatial constraints. It is essentially an enforced encoder fusing the information from the language prompt and the spatial constraints. Connected with the main transformer encoder during training, the realism guidance is trained to correct the signal passed in the corresponding layers.

**Weaknesses:**

- The paper writing is not fluent enough and needs polishing to be easier to follow.
- Given the carefully designed modules, the time efficiency for training is important to evaluate the significance of the proposed method. However, this part is missing in the paper.
- Some important baselines are missing in the experiment sections, such as [1,2]. Adding the full set of published baselines on the benchmarks of HumanML3D and KIT-ML will change the position of the proposed methods highly. Can the authors elaborate more about the comparison with the baselines? Or maybe there is any reason that these baselines are not proper to compare with?
- Some minor writing issues, such as duplicated typo: input -> "input"
- Referring to Figure 5, which is placed in a very late position in the introduction section makes a bad reading flow. You may want to adjust the position of Figure 5 to make it closer to where it is referred to.

Reference:

[1]: "T2M-GPT: Generating Human Motion from Textual Descriptions with Discrete Representations” CVPR 2023

[2]: “Generating Diverse and Natural 3D Human Motions from Text”, CVPR 2022

**Questions:**

See my concerns listed above.

---

> ### Author Response · Authors · 2023-11-21
> **Response to Reviewer gCs4**
>
> Thank you for your time and helpful feedback. We respond below to your questions and comments:
>
> ```
> Q1: The time efficiency for training…
> ```
> **A1**: Thank you for the suggestion to add the training time. It takes **29 hours** to train our model on a single NVIDIA RTX A5000 GPU with batch size of 64 and 250,000 iterations in total.
> As a reference, MDM, PriorDM, and GMD take 72, 11.5, and 39 hours, respectively, according to what is reported in their papers.
> MDM and PriorMDM are trained on a single NVIDIA GeForce RTX 2080 Ti GPU (slower than the A5000 GPU).
> GMD is trained on a single NVIDIA RTX 3090 (roughly comparable with the A5000 GPU).
> Our training efficiency is better than GMD. PriorMDM performs better in training efficiency because it shares the same architecture as MDM and thus can directly finetune the pre-trained weight of MDM (the pre-training time was not included in the reported timing).
> We have added this information in **Table 4 in Appendix A.2** of the revised version.
>
> &nbsp;
>
> ```
> Q2: Some important baselines are missing in the experiment sections … maybe there is any reason that these baselines are not proper to compare with.
> ```
> **A2**: Thank you for sharing these two papers. These baselines [1, 2] are indeed not feasible to compare with because they **cannot** integrate spatial control signals into text-based human motion generation, which is the main focus of our paper.
> The baselines we compared in our paper are all the methods that can integrate the spatial control signals (to our best knowledge). We have compared with
> 1. MDM and PriorMDM: they demonstrate that the inpainting-based methods can be integrated into their pipeline to achieve pelvis-controlling with dense signal in their paper. They can only control the pelvis with the global control signal.
> 2. GMD: it focuses on exactly the same problem as ours but can only control the pelvis.
>
> *[1]: "T2M-GPT: Generating Human Motion from Textual Descriptions with Discrete Representations” CVPR 2023*
> *[2]: “Generating Diverse and Natural 3D Human Motions from Text”, CVPR 2022*
>
> &nbsp;
>
> ```
> Q3: typo Imput -> input
> ```
> **A3**: We believe you meant “imput -> input”. We have revised it to avoid any confusion.
>
> &nbsp;
>
> ```
> Q4: Referring to Figure 5, which is placed in a very late position…
> ```
> **A4**: Thank you for the suggestion. We have removed the reference to Fig. 5 in the introduction section. It is now only referred to in Section 4.2, which is close to its position and should improve the reading flow.
>
> &nbsp;
>
> ```
> Q5: The paper writing is not fluent enough and needs polishing to be easier to follow.
> ```
> **A5**: We have revised the paper to improve the reading flow. We would greatly appreciate it if more insights could be shared on how to further improve it.

---

> > ### Author Response · Authors · 2023-11-22
> > **Follow up with Reviewer gCs4**
> >
> > Dear Reviewer gCs4,
> >
> > Thank you for your detailed review and the valuable feedback! We would love to hear your thoughts about our rebuttal, including whether it sufficiently addresses your concerns and questions. Any feedback is welcome and greatly appreciated!
> >
> > Sincerely,
> > Authors of OmniControl

---

> > ### Comment · Reviewer_gCs4 · 2023-11-22
> > **Feedback to authors' reply**
> >
> > Thanks for your reply.
> >
> > I read your revised version of the paper and the reply carefully. I think the current version is better organized and more self-contained. I appreciate the efforts of adding training time and more details as elaborated.
> >
> > Yes, I intended to indicate the typo "imput->input" ... how could I write typo when indicating a typo :(
> >
> > Thanks for the clarification about the reason of not introducing some related baseline methods in the evaluation.
> >
> > I will be re-considering my rating after the discussion with other reviewers.

---

> > > ### Author Response · Authors · 2023-11-22
> > > **Follow up with Reviewer gCs4**
> > >
> > > Thank you for your thorough feedback. We appreciate your time and consideration in reviewing our work.

---

### Official Review · Reviewer_YHiJ · 2023-10-31

**Soundness:** 3 good
**Presentation:** 3 good
**Contribution:** 3 good
**Rating:** 6
**Confidence:** 4

**Summary:**

The paper introduces OmniControl, a novel method that enhances text-conditioned human motion generation by allowing flexible spatial control across multiple joints, ensuring realistic and coherent movements. The integration of spatial and realism guidance achieves a balance between accuracy and natural motion, demonstrating superior pelvis control and promising outcomes on various joints, marking an advancement in generating constrained, realistic human motions. The commitment to releasing code and model weights further enhances accessibility for future advancements in this field.

**Strengths:**

The paper offers a simple yet effective method to integrate spatial control signals into a text-conditioned human motion generation model based on the diffusion process.
The introduction of realism guidance to refine all joints for generating more coherent motion is commendable.
The evaluation is adequate and comprehensive.

**Weaknesses:**

It would be better if the difference and advantage between the global coordinates and local coordinates could be visualized.
Inference time is higher than MDM and GMD.
The concept in Fig. 4, such as the input process, requires further clarification for better comprehension. In addition, The components of spatial encoder F and the size of output f_n are not explained.
The difference and advantage between the global coordinates and local coordinates were not visually explained.

**Questions:**

In Fig. 7, understanding why higher density leads to higher FID and Foot skating ratio while other factors lead to lower FID and Foot skating ratio is required. Traditionally, higher density in certain contexts can lead to better performance due to increased information or more complex interactions. However, in your case, it seems to be causing a lower performance. Additionally, the paper mentions the Avg. error of MDM and PriorMDM being zero due to the inpainting property. Elaborating on the nature of this property would provide clarity. Moreover, why the proposed methods are with zero error when density is low should be addressed.
In the supplementary video, it would be better if the video demonstrate “Control other joints” can be visualized compared with GMD.
Q: Where is the spatial control signal coming from? Is it given by dataset?

---

> ### Author Response · Authors · 2023-11-21
> **Response to Reviewer YHiJ (1/2)**
>
> Thank you for your thoughtful comments. Please refer to “The General Response to Reviewers” for the reply to the issue of the inference speed. We respond below to your other questions and concerns:
>
> ```
> Q1: The difference and advantage between the global coordinates and local coordinates…
> ```
> **A1**: We show the visualization results with global pose representation in Appendix A.6. We found the model cannot converge and produce collapsed human poses as shown in [Figure 8](https://github.com/OmniControl24/OmniControl24.github.io/blob/main/static/images/figure8.jpg), which do not form the correct motion of a human.
> Recent work InterGen [1] proposes a global representation with bone length loss to enforce skeleton consistency and avoid collapsed human poses. We train MDM [2] with this global representation and bone length loss, and report the performance in [Table 6](https://github.com/OmniControl24/OmniControl24.github.io/blob/main/static/images/table6.jpg). The results show that there is still a significant performance drop in performance when we switch to the global representation for text-based human motion generation. Therefore, global coordinates are not an optimal choice for our task. We added these discussions to **Appendix A.7**.
>
> *[1] Liang, Han, Wenqian Zhang, Wenxuan Li, Jingyi Yu, and Lan Xu. "InterGen: Diffusion-based Multi-human Motion Generation under Complex Interactions." arXiv preprint arXiv:2304.05684 (2023).*
> *[2] Guy Tevet, Sigal Raab, Brian Gordon, Yonatan Shafir, Daniel Cohen-or, and Amit Haim Bermano. “Human motion diffusion model.” In ICLR, 2023.*
>
> &nbsp;
>
> ```
> Q2: The concept in Fig. 4, such as the input process, requires further clarification…
> ```
> **A2**: We have clarified these in **Appendix A.2** in the revised version. We post them here for convenience:
> The input process mainly consists of a CLIP-based textual embedding to encode the text prompt and linear layers to encode the noisy motion. Then the encoded text and noisy motion will be concatenated as the input to the self-attention layers.
> The spatial encoder consists of four linear layers to encode the spatial control signals.
> The size of f_n is (L, B, C), where L = 196 is the sequence length, B is the batch size, and C=512 is the feature dimension.
>
> &nbsp;
>
> ```
> Q3: In Fig. 7, understanding why higher density leads to higher FID and Foot skating ratio…
> ```
> **A3**: Ideally, higher density should lead to better performance if the generated motion can accurately follow the control signal, and make corresponding adjustments to the other joints to make the motion realistic and natural.
> This is not the case for MDM and PriorMDM since they cannot effectively modify whole-body motion according to the input control signal.
> When the density is higher (the constraint is stricter) and other joints are NOT adjusted effectively to compensate for the more rigidity in the control signal, they will produce unnatural results, thus leading to higher FID and higher foot skating ratio.
> On the contrary, GMD and ours, which are specially designed for both text and spatial control signal conditions, can efficiently adjust the whole-body motion and better leverage the context information in the input signal, yielding better performance when the density is higher.
> We have added such discussions for Fig. 7 in **Appendix A.9** of the revised version.
>
> &nbsp;
>
> ```
> Q4: The Avg. error of MDM and PriorMDM is zero due to the inpainting property…
> ```
> **A4**: Inpainting-based methods aim to reconstruct the rest of joint motions based on the given control signals over one or more control joints. The input control signals won't be changed during this process, i.e., the output motion over the control joints remains the same as the input control signal. As a result, the Avg. error is zero.
> We have clarified this point in **Appendix A.9** of the revised version.
>
> &nbsp;
>
> ```
> Q5: Why are the proposed methods with zero error when density is low…
> ```
> **A5**: [Table 10](https://github.com/OmniControl24/OmniControl24.github.io/blob/main/static/images/table10.jpg) in the revised paper shows the full results of the HumanML3D dataset. The Avg. error is not zero when the density is low. The Traj. err. and Loc. err. are sometimes zeros when the density is low since the definitions of these two metrics are not strict.
> Following GMD, Traj. err. (50 cm) is the ratio of unsuccessful trajectories. The unsuccessful trajectories are defined as the trajectories with any keyframe whose location error exceeds a threshold (50 cm). And Loc. err. (50 cm) is the ratio of unsuccessful keyframes whose location error exceeds a threshold (50 cm). When the density is low (e.g., only have spatial control signal in one keyframe), it is easier for all samples to meet this threshold and thus achieve zero errors. We have added this explanation to **Appendix A.12** of the revised version.

---

> ### Author Response · Authors · 2023-11-21
> **Response to Reviewer YHiJ (2/2)**
>
> ```
> Q6: In the supplementary video “Control other joints”, it would be better if compared with GMD…
> ```
> **A6**: Unfortunately, GMD can only control the pelvis as discussed in the introduction (paragraph 2), so this comparison is not feasible.
>
> &nbsp;
>
> ```
> Q7: Where is the spatial control signal coming from…
> ```
> **A7**: In both training and evaluation, all models are provided with ground-truth trajectories as the spatial control signals. In the visualizations or video demos, the spatial control signals are manually designed. More details can be found in **Appendix A.11**.

---

> > ### Author Response · Authors · 2023-11-22
> > **Follow up with Reviewer YHiJ**
> >
> > Dear Reviewer YHiJ,
> >
> > Thank you for your detailed review and the valuable feedback! We would love to hear your thoughts about our rebuttal, including whether it sufficiently addresses your concerns and questions. Any feedback is welcome and greatly appreciated!
> >
> > Sincerely,
> > Authors of OmniControl

---

### Official Review · Reviewer_EL5t · 2023-10-31

**Soundness:** 3 good
**Presentation:** 3 good
**Contribution:** 4 excellent
**Rating:** 8
**Confidence:** 3

**Summary:**

The paper presents a unified approach that can support controlling any joint at any time for text-driven human motion synthesis. The core design is to integrate both spatial and realism guidances to keep the generated motion faithful to the control signals while improving its reality and naturalness. Experiments show that the proposed method outperforms baselines in terms of control accuracy, motion reality, and motion diversity.

**Strengths:**

* (1) The method is the first to control any joint at any time for human motion synthesis, which can improve the flexibility of motion generation tasks and potentially benefit downstream applications such as generating human motion on different terrains.

* (2) The method design is clear and reasonable.

* (3) Experiments demonstrate the effectiveness of the proposed method.

* (4) The analysis for method ablations is solid.

* (5) The paper is well-organized and easy to follow.

**Weaknesses:**

* (1) The inference speed for the proposed method is much lower than baselines, which could potentially impede the method to apply to a large amount of data.

* (2) In the third column of Figure 1, the authors show that the method can support a combination of control signals from different joints. However, the paper lacks quantitative analysis to further examine its performance.

**Questions:**

I wonder whether the proposed method can support motion editing where after a motion is generated, control signals can be edited and can further adjust the motion to be not only close to the previous one but also faithful to the new control signals.

---

> ### Author Response · Authors · 2023-11-21
> **Response to Reviewer EL5t**
>
> Thank you for your time and helpful feedback. Please refer to “The General Response to Reviewers” for the reply to the issue of the inference speed. We respond below to your other comments and questions.
>
> ```
> Q1: Lack of quantitative analysis of the combination of control signals from different joints…
> ```
> **A1**: Thank you for the suggestion! We have added such quantitative results of the combination of control signals from different joints, which are inserted into **the last row of Table 1**. We have also added more discussions in **Appendix A.4**.
> We also post the results here:
> |              | FID ↓ | R Precision (top 3) ↑ | Diversity-9.503→ | Foot sliding ratio↓ | Traj. err. ↓ (50 cm) | Loc. err. ↓ (50 cm) | Avg. err. ↓ |
> |--------------|----------------|----------------------------------|------------------|---------------------|----------------------|---------------------|-------------|
> | Single Joint | 0.310          | 0.693                            | 9.502            | 0.0608              | 0.0617               | 0.0107              | 0.0404      |
> | **Combination**  | 0.624          | 0.672                            | 9.016            | 0.0874              | 0.2147               | 0.0265              | 0.0766      |
>
> There are 57 possible combinations for six types of joints. Since running an evaluation for each of them is costly, it's impractical to evaluate all the combinations. Instead, we randomly sample one possible combination for each motion sequence for evaluation.
> The performance is lower compared to the single-joint control (not an apple-to-apple comparison though as the ground-truths are different). Nevertheless, the results show that controlling multiple joints is harder than a single one due to the increased degrees of freedom.
>
> &nbsp;
>
> ```
> Q2: Whether the proposed method can support motion editing where after a motion is generated…
> ```
> **A2**: It depends on whether we can re-run our pipeline.
> If yes, we can achieve this by following these steps: (1) extract the global trajectories of all the joints from the already generated motion; (2) Edit the control signals; (3) Combine the edited control signal and the global trajectories of rest joints from generated motion as the new spatial control signal; and (4) regenerate the motion using the new control signal.
> The edited control signal ensures the motion can be faithful to the new edition and the global trajectories make the already generated motion close to the previous one. If not, the motion cannot be edited without re-running our pipeline as far as we know. That would be interesting future work.
>
> We would greatly appreciate it if you could let us know if these can address your question.

---

> > ### Author Response · Authors · 2023-11-22
> > **Follow up with Reviewer EL5t**
> >
> > Dear Reviewer EL5t,
> >
> > Thank you for your detailed review and the valuable feedback! We would love to hear your thoughts about our rebuttal, including whether it sufficiently addresses your concerns and questions. Any feedback is welcome and greatly appreciated!
> >
> > Sincerely,
> > Authors of OmniControl

---

> > ### Comment · Reviewer_EL5t · 2023-11-22
> >
> > I appreciate the author's response. It has addressed my concerns.

---

> > > ### Author Response · Authors · 2023-11-22
> > > **Follow up with Reviewer EL5t**
> > >
> > > Thank you for your feedback. We appreciate your time and consideration in reviewing our work!

---

### Author Response · Authors · 2023-11-21
**The General Response to Reviewers**

We thank the reviewers for their thorough and thoughtful comments. We are glad to see that the design of OmniControl is recognized as “**clear and reasonable**” (EL5t), the paper is “**well-organized and easy to follow**.” (EL5t), “**the introduction of realism guidance … is commendable**" (YHiJ), “**the evaluation is adequate and comprehensive**” (YHiJ), “**the paper considers the novel task**” (4D6Y), “**experiments … show promising results**” (4D6Y), and “**the accompanying visualizations are of high quality**”(4D6Y).

In the updated version of the paper, changes are marked in yellow. The changes to the paper and supplementary videos following your suggestions are summarized as follows:
1. Added a row to Table 1 to evaluate the model with the combination of control signals for different joints and added more discussions in Appendix A.4.
2. Added more discussions about the relative and global human representation in Appendix A.7.
3. Added more implementation details in Appendix A.2.
4. Added more discussions about Fig. 7 in Appendix A.9.
5. Added more results discussion in Appendix A.12.
6. Discussed the source of spatial control signals in Appendix A.11.
7. Revised “imput -> input”.
8. Removed the reference to Fig. 5 in the introduction section.
9. Fixed the minor issues in the citation format.
10. Added visualization results with extremely unnatural spatial control signals to the supplementary video [(from 4:41 to 5:23)](https://youtu.be/29M-Shs1Orw?t=284).
11. Provided visualization results with only a textual description or trajectory as the prompt in the supplementary video [(From 5:23 to 5:45)](https://youtu.be/29M-Shs1Orw?t=325).

**We also provide more video results on this [anonymous website](https://omnicontrol24.github.io/)**.  We greatly appreciate your insightful reviews. The latest revision has become notably more comprehensive and thorough, largely due to your valuable feedback!

***
To reviewers **EL5t and YHiJ**, we address your shared comment about the inference time as follows.
```
Q1: The inference speed for the proposed method is much lower than baselines.
```
**A1**: Indeed, our approach exhibits a longer inference time compared to MDM (ours: 121s vs. MDM: 39s), but it is comparable to that of GMD (110s). This increase in computation time is offset by the ability of flexible and fine-grained control for human motion generation. At the same time, we see significant potential for enhancing the computational efficiency of our method. Recent advancements in diffusion models [1,2,3,4] present promising avenues for this, and exploring these will be an exciting direction for future work. This could potentially enable our approach to handle large datasets more effectively.

*[1] Salimans, Tim, and Jonathan Ho. "Progressive distillation for fast sampling of diffusion models." arXiv preprint arXiv:2202.00512 (2022).*
*[2] Lyu, Zhaoyang, Xudong Xu, Ceyuan Yang, Dahua Lin, and Bo Dai. "Accelerating diffusion models via early stop of the diffusion process." arXiv preprint arXiv:2205.12524 (2022).*
*[3] Lu, Cheng, Yuhao Zhou, Fan Bao, Jianfei Chen, Chongxuan Li, and Jun Zhu. "Dpm-solver: A fast ode solver for diffusion probabilistic model sampling in around 10 steps." Advances in Neural Information Processing Systems 35 (2022): 5775-5787.*
*[4] Song, Yang, et al. "Consistency Models." arXiv preprint arXiv:2303.01469 (2023).*

***
In addition, we address individual comments and questions by commenting on your reviews.

---

### Meta-Review · Area_Chair_f3Ya · 2023-12-11

**Metareview:**

The paper proposes a text-conditioned human motion generation method (OmniControl) that makes it possible to incorporate spatial control signals over different joints. The method employs spatial and realism guidance in an effort to achieve a balance between motion that is both accurate and natural. Experiments demonstrate that the method outperforms contemporary motion generation methods that employ pelvis control in terms of control accuracy, the realism of the resulting motion, and motion diversity.

The paper was reviewed by four referees who's opinions on the paper largely agree. As several reviewers point out, the ability to incorporate control signals over different joints is a novel contribution that is shown to provide several advantages over existing pelvis control-based generation methods. The underlying method is principled and clearly presented. One limitation of the method is the relatively high cost of inference, which several reviewers point out and the authors acknowledge in their response. There were some initial questions/concerns about the omission of relevant baselines and the lack of training details, both of which were resolved during the author-reviewer discussion phase. The result is a paper that provides a nice contribution to the community.

NOTE: Reviewer 4D6Y did not reply to the authors despite several attempts by the authors and AC to engage the reviewer. In light of this as well as the fact that some of the concerns that they raised lack merit, the AC placed less weight on their review when making a recommendation.

**Justification For Why Not Higher Score:**

I don't think that the contributions are sufficiently significant to warrant a spotlight.

**Justification For Why Not Lower Score:**

The significance of the paper's contributions is sufficient.

---

### Decision · Program_Chairs · 2024-01-16

Accept (poster)